# Molecularly-porous ultrathin membranes for highly selective organic solvent nanofiltration

Tiefan Huang[1,2], Basem A. Moosa[3], Phuong Hoang[3], Jiangtao Liu [1], Stefan Chisca[1], Gengwu Zhang[3], Mram AlYami[3], Niveen M. Khashab [3✉] & Suzana P. Nunes [1✉]

Engineering membranes for molecular separation in organic solvents is still a big challenge. When the selectivity increases, the permeability tends to drastically decrease, increasing the energy demands for the separation process. Ideally, organic solvent nanofiltration membranes should be thin to enhance the permeant transport, have a well-tailored nanoporosity and high stability in harsh solvents. Here, we introduce a trianglamine macrocycle as a molecular building block for cross-linked membranes, prepared by facile interfacial polymerization, for high-performance selective separations. The membranes were prepared via a two-in-one strategy, enabled by the amine macrocycle, by simultaneously reducing the thickness of the thin-film layers (<10 nm) and introducing permanent intrinsic porosity within the membrane (6.3 Å). This translates into a superior separation performance for nanofiltration operation, both in polar and apolar solvents. The hyper-cross-linked network significantly improved the stability in various organic solvents, while the amine host macrocycle provided specific size and charge molecular recognition for selective guest molecules separation. By employing easily customized molecular hosts in ultrathin membranes, we can significantly tailor the selectivity on-demand without compromising the overall permeability of the system.

[1] Nanostructured Polymeric Membranes Laboratory, Advanced Membranes and Porous Materials Center, Biological and Environmental Science and Engineering Division (BESE), King Abdullah University of Science and Technology (KAUST), Thuwal 23955-6900, Saudi Arabia. [2] Functional Membrane Materials Engineering Research Center of Hunan Province, School of Chemistry and Chemical Engineering, Hunan University of Science and Technology, 411201 Xiangtan, China. [3] Smart Hybrid Materials (SHMs) Laboratory, Advanced Membranes and Porous Materials Center, Physical Sciences and Engineering Division (PSE), King Abdullah University of Science and Technology (KAUST), Thuwal 23955-6900, Saudi Arabia. ✉email: niveen.khashab@kaust.edu.sa; suzana.nunes@kaust.edu.sa

Separation, recovery, and disposal of organic solvents have always been a challenge for the petroleum, chemical, and pharmaceutical industries. Traditional chemical engineering processes, such as distillation, adsorption, and extraction, demand high capital and operating cost, which consume large amounts of energy and impose a heavy environmental footprint. An alternative to these technologies is organic solvent nanofiltration (OSN), which usually requires less energy and is more environmentally friendly[1,2]. However, similar to other membrane-based processes, nanofiltration is heavily plagued by the trade-off between permeability and selectivity[3]. The effective paths for molecular transport in most membranes are not strictly uniform. The precise tuning of the pore size and size distribution to promote high selectivity control at the nanoscale is still in high demand. New strategies providing permanent and uniform porosity are critical to enabling more effective molecular separations[4–6].

Interfacial polymerization has been used for decades for thin-film composite (TFC) membrane fabrication, leading to high permeance[7–10]. It can generate in situ cross-linked polymer layers with demonstrated stability in water and organic solvents. A major drawback of these composite membranes has been the poor selectivity in the separation of molecules of similar size in the nanofiltration range. New monomeric structures are under investigation to improve the overall performance (permeance and selectivity) of TFC membranes[11]. Related approaches consider the integration of well-defined crystalline microporous materials such as metal−organic frameworks (MOFs) and covalent organic frameworks (COFs) for selective separations[12–16].

However, to fabricate such crystalline membranes free from defects is difficult. A common and simple strategy is the incorporation of microporous materials as fillers in the polymer matrix, forming mixed-matrix membranes (MMMs)[17]. These additives can in principle provide intrinsic molecular transport channels to facilitate the solvent permeability. The main drawback of discrete additives embedded within a polymer matrix is the poor adhesion and aggregation, which can induce the formation of nonselective voids and decrease the separation performance.

Macrocyclic molecules with intrinsic microporosity[18] have been employed as recognition units for selective separation in active layers of nanofiltration and molecularly mixed composite membranes preparation[19–22]. One approach has been the incorporation of charged macrocycles, such as sufocalix[4]arene into films by ionic interaction with polyelectrolytes or by electron donor–acceptor interaction in polymer networks[23,24]. Cyclodextrins have been incorporated in different forms for membrane preparation aiming at the removal of organic pollutants, for instance, blended with film-forming polymers[25]. A more successful approach has been effectively linking macrocycles, such as cyclodextrin, to secure stability and avoid leaching out during operation. They have been fixed, for instance, through host–guest interactions with adamantylamine, used as co-monomer in an interfacial polymerization reaction[26] or tethered to plasma modified surfaces[27]. Cyclodextrins have been cross-linked in polymeric backbones for the rapid removal of organic pollutants for use as adsorbents[28]. When targeting membranes for liquid separation stability, high permeance and selectivity are required, and integrating functional macrocycles as a reactive monomer in interfacial polymerization has been demonstrated as one of the most effective approaches for nanofiltration[29,30]. Considering the rigid and well-defined hollow cavity of cyclodextrins, the membrane was endowed with fast solvents permeance and shape selectivity for molecules. More recently, Noria, a double cyclic ladder-type oligomer, has been successfully used as a recognition unit in selective nanofiltration membranes[20]. However, covalently

bonding a macrocyclic host building block to a polymeric matrix has been limited due to solubility considerations, which drastically affected the overall cross-linking. Moreover, employing macrocycles that could be easily functionalized and scaled up is pivotal for the industrial translations of these membranes.

Herein, a two-in-one strategy to prepare molecularly porous cross-linked membranes (MPCM) is simultaneously introduced, reducing the thickness and introducing micro-isoporosity within the membrane. This ultimately should furnish the MPCMs with both high permeance and sharp selectivity. Trianglamines[31], with an intrinsic microporous structure, was chosen as a monomer for the membrane preparation via interfacial polymerization.

## Results

**Construction of MPCM.** Trianglamine was synthesized as previously reported[31]. Trianglamine was selected as a monomer for membrane fabrication based on its size, functionality, and synthesis (yield ≥ 90%). A further important advantage is a possibility of easily tuning the pore size by expanding the length of the macrocycle linker, keeping analogous chemistry, or by constricting the effective channel with the addition of side groups. In this way, the procedure demonstrated here with trianglamine could be easily extended to other macrocycles, supplying a portfolio of membranes for tailored molecular separations, just by substituting the main building block. We conducted the interfacial polymerization of trianglamine solubilized in water, in contact with an organic phase, containing acyl chloride. An ultrathin MPCM was obtained. The six reactive amino groups per trianglamine molecules led to the formation of a high cross-linking density with demonstrated excellent performance as a membrane for molecular separation in an organic solvent medium.

A certain amount of trianglamine (2 wt%) was dispersed in water, and the pH was adjusted by adding HCl to have a transparent solution (Supplementary Figs. 1 and 2). Terephthaloyl chloride (TPC) was added to the organic phase (Fig. 1). The scanning electron microscopy (SEM) image of the interfacially polymerized layer on polyacrylonitrile (PAN) support shows a continuous and smooth surface without any identifiable pinhole or crack (Fig. 2a and Supplementary Figs. 3–5). The thickness could not be precisely quantified in the cross-sectional SEM images, since the boundary between the supports and the ultrathin nanofilms could be hardly distinguished (Supplementary Figs. 6 and 7). Therefore, a freestanding cross-linked trianglamine nanofilm was prepared under analogous conditions, illustrated in Fig. 2b. The film was transferred onto another substrate, as described in Supplementary Fig. 8. Figure 2c–e shows the atomic force microscopy (AFM) images of the film on a silicon wafer. The height profile reveals that the thinnest film is around 3.5 nm (Fig. 2d). The thickness of the film is tunable depending on the synthesis condition (Supplementary Fig. 9). By increasing the reaction time from 10 s to 5 min, the thickness of the film increased from 3.5 to 10 nm. A further time increase did not lead to even thicker films. The same trend was observed for the film surface roughness. Nevertheless, all films have a root-mean-square roughness below 1 nm, indicating the highly smooth and flat morphologies of the films, which agrees with the SEM observations. The freestanding film with 10 min reaction time covering the alumina support is shown in Fig. 2g–i. The thickness of the film could be clearly measured from the cross-sectional image as being around 13 nm. Considering that the iridium coating for the SEM sample preparation is 3 nm thick, the thickness of the polymerized layer measured by SEM is consistent with the AFM result (Supplementary Fig. 9a). The MPCM nanofilm has an even and flat morphology. The porous alumina

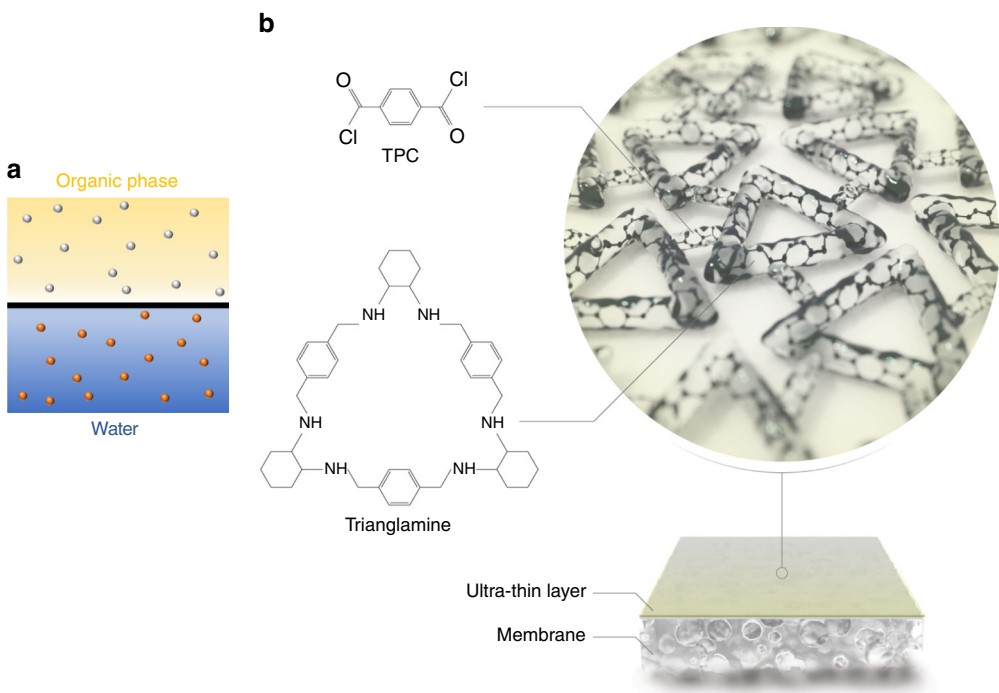

**Fig. 1 Interfacial polymerization of MPCM. a** TPC (gray dots) dissolved in an organic phase, trianglamine (red dots) in an aqueous phase, reacting to form **b** the cross-linked network of MPCM.

support under the film is visible, indicating that it is extremely thin (Fig. 2h). The high-resolution TEM (HR-TEM) images reveal that the film thickness is around 17–20 nm and without any particular order or crystallinity (Supplementary Fig. 10). The lack of crystallinity was confirmed by wide-angle X-ray diffraction (XRD) (Supplementary Fig. 11).

The formation of defect-free hyper-cross-linked trianglamine layers thinner than 20 nm for OSN application using the reported strategy with no additional steps, like applying a sacrificial interlayer, is advantageous in terms of selectivity, permeance, and potential for scaling-up. We then investigated the mechanism of the film formation between the aqueous and organic phases. Since the polymer film formation takes place more in the organic side of the interface, we speculated that a high solubility and diffusivity of the aqueous monomer into the organic phase could expanse polymerization zone and consequently increase the film thickness, and vice versa. As shown in Supplementary Fig. 12, the diffusivity of trianglamine into the organic phase is extremely slow. This can be ascribed to the ionic feature of the trianglamine monomer in the aqueous phase and to its large size compared to classical amines, such as m-phenylenediamine (MPD), used for interfacial polymerization. As a comparison, MPD shows much higher solubility and faster diffusivity under the same condition. The reaction interface with trianglamine will be less susceptible to convection, explaining the smoothness of the final layer.

**Membrane chemical characterization.** The chemical structure as polyamide of the MPCM was verified by ATR-FTIR spectra (Fig. 3a and Supplementary Fig. 13). After polymerization, the stretching bands of O=C–Cl and –NH– groups, which appeared at 1760 cm$^{-1}$ and 3290 cm$^{-1}$ in the spectra of TPC and trianglamine, respectively, are greatly attenuated in the film. Simultaneously, a new strong stretching frequency band appears at 1623 cm$^{-1}$, which can be attributed to the –C=O bonds from the secondary amide group[32]. These results clearly support the formation of the polyamide film by an amide reaction between TPC

and trianglamine. Additional less pronounced changes in the spectrum after cross-linking appear at 750, 1260, and 1400 cm$^{-1}$, which could be assigned to O=C–N bending in-plane, C–N stretching, and a less specific absorption for polyamides, respectively. Besides these peaks, the overall similarity of the spectra of the network and the pristine trianglamine suggested that the macrocycle skeleton remains intact. To further quantify the cross-linking degree of the film, X-ray photoelectron spectroscopy (XPS) was carried out (Fig. 3b–d and Supplementary Figs. 14 and 15 and Supplementary Table 1). Compared to the PAN support, the O/N ratio of the coated membrane increased from 0.04 to 0.7. The higher O/N ratio suggests that the top surface of the PAN support is covered by the MPCM nanofilm layer. However, due to the interference of the PAN support in the XPS spectrum of the selective layer, we cannot obtain precise XPS information of the pure MPCM on the PAN membrane. Therefore, a freestanding MPCM nanofilm was collected from the interface between the aqueous and organic phases to further clarify its chemical structure. As a comparison, the trianglamine monomer was also examined. As for the trianglamine, there is the main peak at 398.8 eV in the N1s spectrum, corresponding to the –NH– (Supplementary Fig. 14b). After polymerization, the main peak shifted to 399.7 eV (Fig. 3d), which is assigned to the N–C=O, and the original peak of –NH– largely decreased[20,25]. The percentage of the reacted –NH– group per trianglamine molecule could be calculated based on the ratio between the peak of N–C=O (399.7 eV) and the whole signal, revealing that 78% of the –NH– groups reacted with TPC. Because of the possible steric hindrance, not all –NH– groups have been consumed in the reaction. The deconvolution of the C 1s narrow scan spectrum identified five-component peaks: C=C (284.6 eV), C–C (285.1 eV), C–N (285.9 eV), N–C=O (287.8 eV), and COO (288.9 eV)[8,33] (Fig. 3c). Among them, the N–C=O and COO are only related to the TPC segments. There would be two potential reasons for these changes. During the polymerization reaction, either each TPC cross-linked two adjacent trianglamines, producing two N–C=O

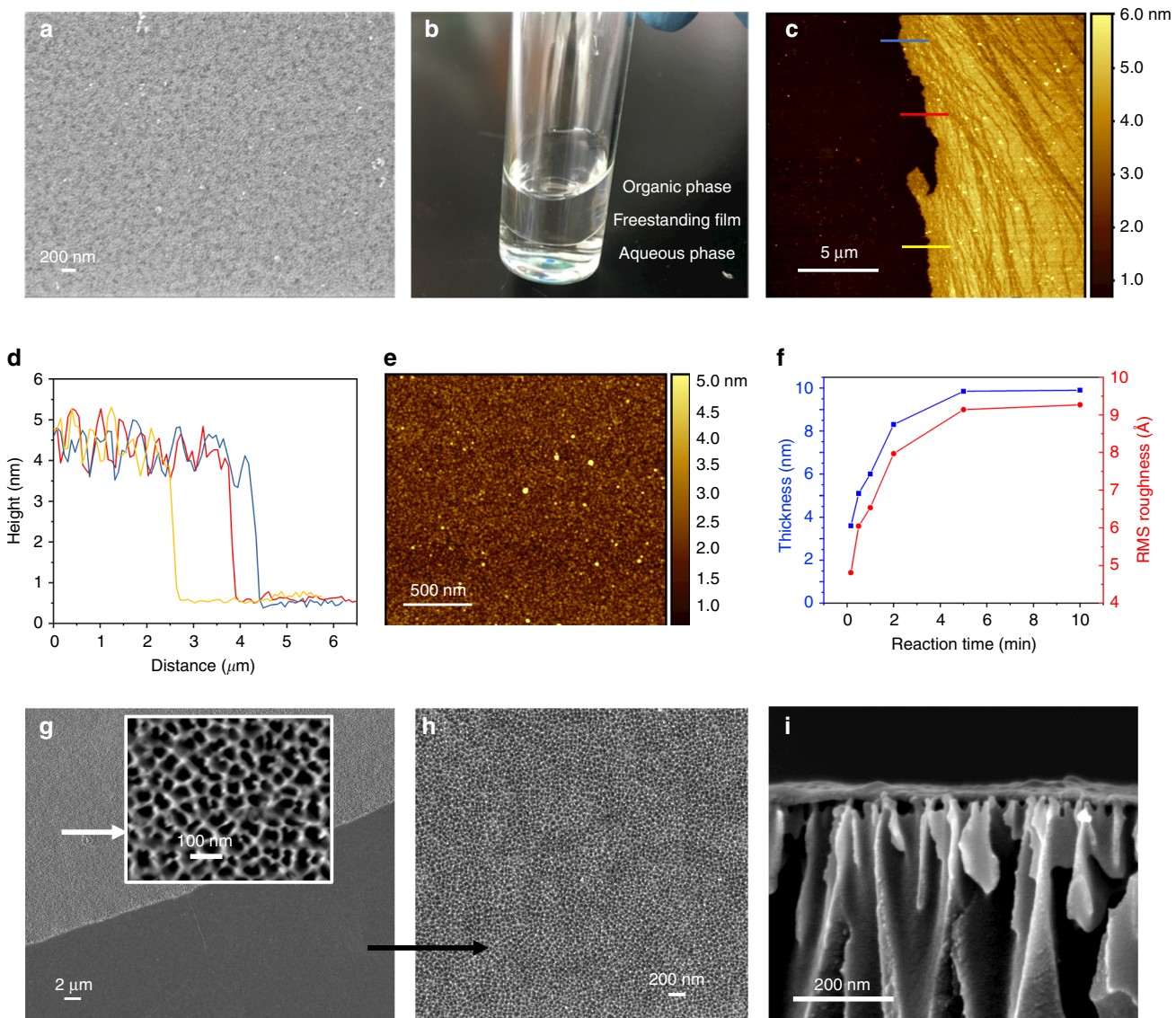

**Fig. 2 Morphologies of hyper-cross-linked MPCM nanofilms. a** SEM top-view image of an MPCM/PAN composite membrane. **b** Incipient MPCM nanofilm at the free interface. **c**–**e** AFM image, height profiles, and surface morphology of the MPCM nanofilm prepared with a reaction time of 10 s on silica wafer (roughness profiles in **d** along the lines shown in **c**). **f** Thickness increase with the reaction time. **g** SEM top-view image of the MPCM nanofilm formed with a reaction time of 10 min on an alumina support, inlet: high magnification image of the bare alumina support. **h** SEM image of the MPCM nanofilm on an alumina support. **i** SEM cross-section image of the MPCM nanofilm covering the alumina support.

groups, or it just attached to one trianglamine without cross-linking, producing one N–C=O group and one COO. Based on the peak ratio of N–C=O and COO, we can calculate that each trianglamine is covalently linked with the other four trianglamines via TPC bridging, and the trianglamine content is estimated to be around 60 wt% in the membrane (estimation detail in Supplementary Information). Such hyper-cross-linking polymerization makes the film dense and highly stable. As a result, the films have excellent stability in a wide range of solvents (Supplementary Fig. 16). The TGA measurement demonstrates their good thermal stability up to 300 °C (Fig. 3e). The sharp weight decrease ranging from 300 to 420 °C corresponds to the trianglamine moiety decomposition. Based on the curve, we can estimate the trianglamine content in the film as being about 62 wt%, which is quite consistent with the result of the XPS evaluation.

The hydrophobicity of MPCM was confirmed, as the contact angle is close to 90° (Supplementary Fig. 17). The PAN support has a low water contact angle. Furthermore, due to the porous

structure, the water droplet gradually infiltrates the support. The trianglamine cross-linked layer has a high proportion of nonpolar groups, i.e., cyclohexyl, coming from the trianglamine moiety of the top-layer and this is reflected in larger water contact angles. Consequently, the membranes show good compatibility for apolar solvents like hexane (Supplementary Fig. 18). Furthermore, due to the presence of the trianglamine layer, there is no detectable water penetration into the membrane, suggesting the formation of a continuous and dense film on the top of PAN support.

**Nanofiltration performance of MPCM.** We have tested the OSN performances of the membranes under certain pressure at 25 °C. Ultrahigh liquid permeance may be achieved in ultrathin membranes because of the short transport path. To verify this further, the membrane permeance with different interfacial polymerization time was tested. As shown in Fig. 4a, the methanol

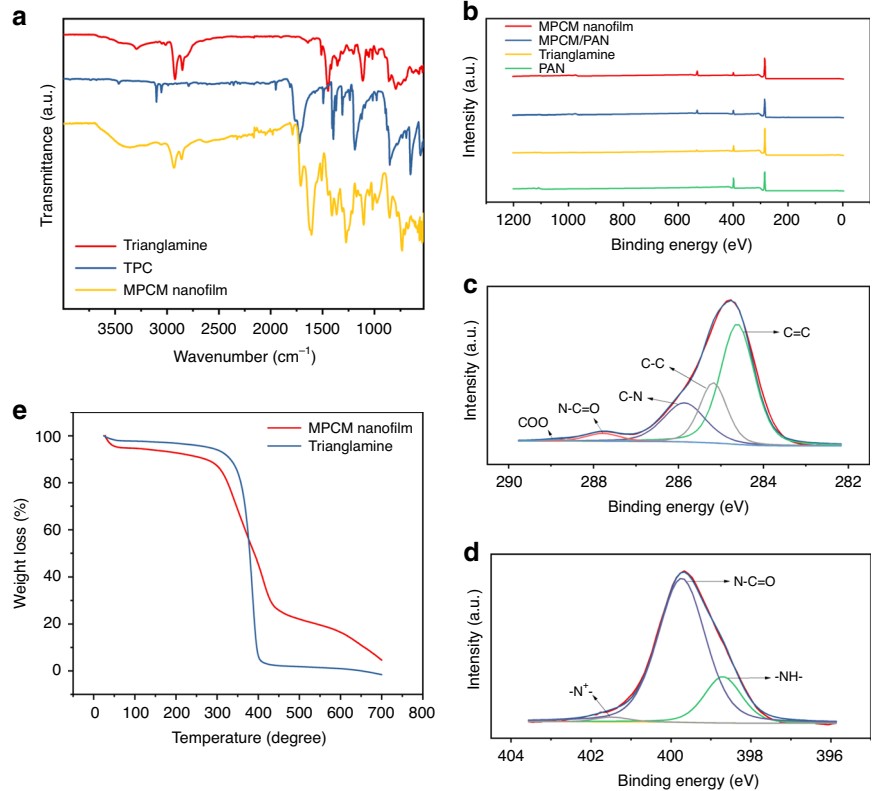

**Fig. 3 Chemical analysis of MPCM. a** ATR-FTIR spectra of trianglamine (in red), TPC (in blue), and MPCM (in yellow). **b** XPS survey spectra of PAN support (in green), trianglamine (in yellow), MPCM/PAN composite membrane (in blue), and MPCM nanofilm (in red) and PAN support (in green). **c** C 1s spectra, and **d** N1s spectra of the MPCM. **e** TGA curves of pristine trianglamine and MPCM.

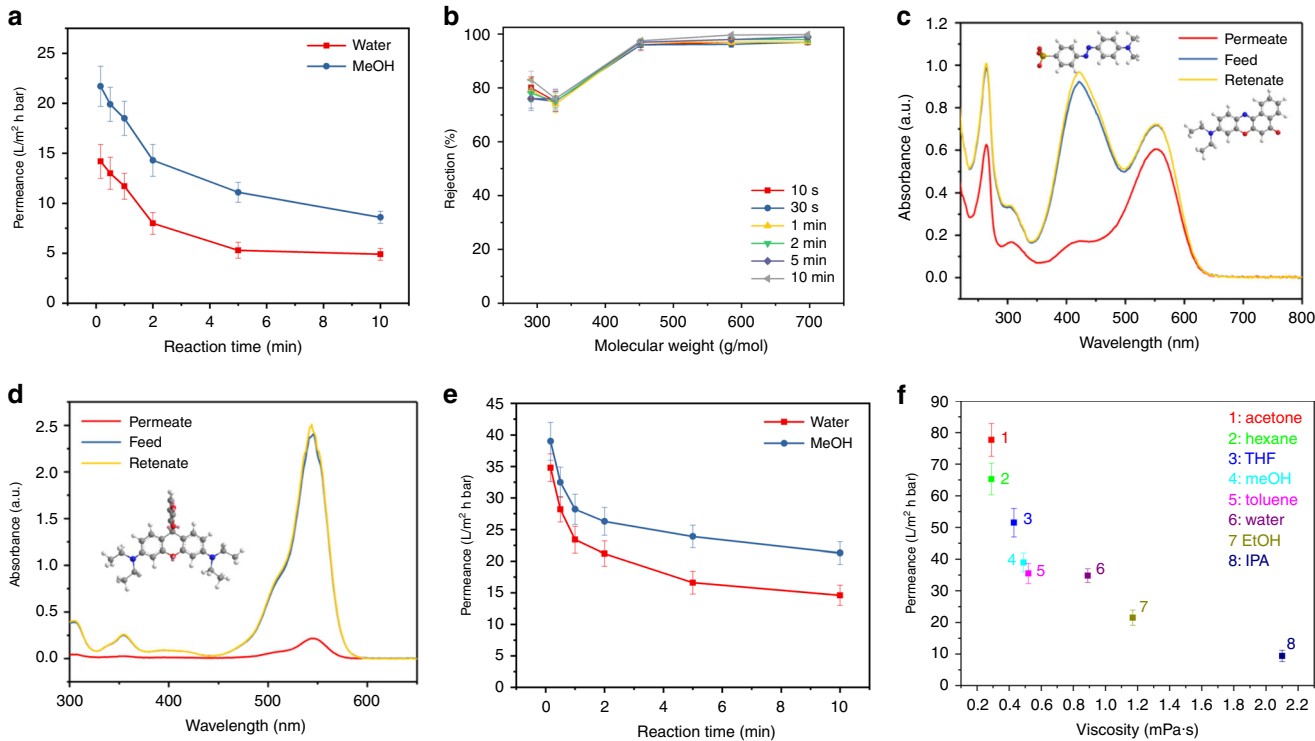

**Fig. 4 Nanofiltration performance. a** Permeance and **b** membrane selectivity for various dye molecules by the MPCM prepared with different reaction time. **c** UV–vis absorption spectra of the mixed dyes (methyl orange and Nile red) in methanol permeation through the MPCM. **d** UV–vis absorption spectra of the rhodamine B base in methanol permeation through the MPCM. (**a–f** membrane prepared with 2 wt% trianglamine). **e** Permeance of the membranes prepared with a 1 wt% aqueous phase concentration. **f** Solvents permeance through the membrane versus the solvent viscosity for the membrane prepared with 1 wt% trianglamine. All error bars represent the standard deviation for at least three independent measurements.

permeance decrease from 22 to 8.6 L m$^{-2}$ h$^{-1}$ bar$^{-1}$ and water permeance from 14 to 5 L m$^{-2}$ h$^{-1}$ bar$^{-1}$ with the reaction time increasing from 10 s to 10 min. More specifically, the permeance sharply decreased from 10 s to 5 min. This follows the trend of the trianglamine-polyamide film thickness, as demonstrated by AFM. Figure 4b shows the membrane selectivity for a range of dye molecules (Supplementary Table 2). Interestingly, the rejection of the membranes is not largely affected by the reaction time, implying that the dense separating layer can be generated even in a 10 s reaction. This is attributed to the high reactivity of the trianglamine monomer and the effective sieving by the formed hyper-cross-linked network structure, greatly retaining the molecules going through. As a result, all membranes manifest a high rejection (>95%) for dyes with a molecular weight larger than 450 g mol$^{-1}$. For the lower molecular weight dyes, such as methyl orange, the rejection was around 80%. In addition to the high permeance and rejection, the membranes also show superior chemical stability. After filtration with a various organic solvent, the membrane can still keep its performance by demonstrating 96% and 83% rejection for orange G (OG) and methyl orange (MO), respectively (Supplementary Fig. 19), demonstrating its outstanding organic solvents resistance. The long-term stability of the membrane was confirmed by testing methanol, acetone, toluene, as well as dye solution of high concentration for more than 48 h. No significant decrease of permeance and rejection was observed (Supplementary Fig. 20). The flux of the membrane was found to be linearly proportional to the applied pressure gradient ($\Delta P$), reflecting that the microporosity within the separating layers remained unchanged with increasing compression (Supplementary Fig. 21).

Significantly, the membrane is charge-selective for molecules of similar size. To illustrate it, the separation of neutral and anionic molecules with similar molecular weight was conducted. Figure 4c shows the separation results performed with mixed solutions of MO (negative, 327 g mol$^{-1}$) and Nile Red (NR) (neutral, 318 g mol$^{-1}$). The neutral NR can freely pass through the membrane, whereas the negatively charged MO was largely retained. The separation of cationic molecules, such as methylene blue (320 g mol$^{-1}$) has also been tested, and low rejection was observed (Supplementary Fig. 22). These results can be attributed to the unique electrostatic property of the cavity of trianglamine and the overall negative charge of the membrane surface. As simulated in Supplementary Fig. 23, the cavity of trianglamine is electron-rich, which could prevent negatively charged molecules from going through it. Considering the high content of trianglamine in the film, the pathway for negatively charged molecules is limited, consequently resulting in high rejection. The membranes possess not only a charge selectivity but also a strict size selectivity, because of the rigid macrocycle structure, densely integrated in the selective layer. Neutral molecules, such as rhodamine B base and NR, are clearly rejected by size. Rhodamine is a bulkier molecule (442 g mol$^{-1}$) than NR (318 g mol$^{-1}$), and has a much higher rejection (91%) (Fig. 4d).

Higher permeance can be obtained by decreasing the trianglamine concentration in the aqueous phase during the membrane preparation. When 1% trianglamine was adopted as a concentration in the aqueous phase while keeping other membrane fabrication parameters the same, the permeance of membranes was doubled, without deterioration of the membrane selectivity (Fig. 4e). For example, with 10 s of reaction time, the methanol permeance reaches 39 L m$^{-2}$ h$^{-1}$ bar$^{-1}$ and the water permeance 35 L m$^{-2}$ h$^{-1}$ bar$^{-1}$. All membranes are able to reject more than 90% of dyes with molecular weight 450 g mol$^{-1}$ or larger (Supplementary Fig. 24). This is evidence that they outperform the majority of the state-of-the-art membranes, including the commercial ones (Supplementary Table 3). Unlike the

reaction time, the monomer concentration heavily affects the membrane performance. A low concentration of TPC leads to high permeance but low rejection, possibly due to defects in the film. For example, at a TPC concentration of 0.0125%, permeances of 77.6 and 57.4 L m$^{-2}$ h$^{-1}$ bar$^{-1}$ were achieved for methanol and water, respectively, along with a low dye rejection (Supplementary Fig. 25). Once the TPC concentration increases to 0.1%, the membrane performance remains practically stable, indicating that a defect-free film can be formed above this concentration.

**Interfacial polymerization with fragmented trianglamine**. To investigate how important the membrane architecture is, a monomer analogous to trianglamine but without the macrocycle structure was used for the membrane preparation via the same procedure. This monomer is a linear fragment of trianglamine with two secondary amino groups, which can lead to practically the same chemistry as the MPCM, but without the intrinsic porosity provided by the macrocycles. Membranes prepared with this fragment instead of trianglamine had a congo red rejection lower than 20%, even when a prolonged reaction time up to 6 h was applied (Supplementary Fig. 26). This can be attributed to the fact that the permeation paths, in this case, are relatively dynamic and sensitive to the relaxation of chains during operation in organic solvents. Since there are only two amino-functional groups per fragment to react with two available acyl chloride groups per TPC molecule, linear chains are formed during the interfacial polymerization, instead of a rigid cross-linked network of trianglamines. The membrane selective layer is therefore constituted by physically entangled polymer chains, and the permeation paths are provided by the polymer interchain distances. Compared to the polymerized trianglamine, the lack of chemical cross-linking of these linear chains could be responsible for lower membrane stability in long-term experiments.

To further evaluate the potential of MPCM for OSN applications, the permeation of membranes prepared with 1% trianglamine in the aqueous phase and reaction time 10 s was measured for various organic solvents (Supplementary Table 4). For molecular-selective nanofiltration membranes, both the thermodynamic interaction with the permeant and its size can influence the transport. In contrast to that observed for traditional polyamide composite membranes, fabricated from TMC and MPD, which are only permeable to polar organic solvents, but not to apolar ones, all tested solvents had a fast transport through the trianglamine membranes (Fig. 4f). This could be attributed to the presence of both hydrophilic (amide and carboxyl) and hydrophobic (cyclohexyl) groups in the MPCM network structure. Besides that, the ultralow thickness and permanently interconnected nano-isoporosity contribute to the high permeance. Nevertheless, the solvent permeance linearly increases in the inverse proportion to the viscosity (Supplementary Fig. 27). The solvent with the lowest viscosity has the highest permeance. For example, acetone, with a viscosity of 0.29 mPa s, had permeance of ~77.7 L m$^{-2}$ h$^{-1}$ bar$^{-1}$, whereas for water it was 35 L m$^{-2}$ h$^{-1}$ bar$^{-1}$, despite the kinetic diameter being almost two times smaller than that of acetone. Isopropanol, with a viscosity of 2.1 mPa s, had the lowest permeance (~9.4 L m$^{-2}$ h$^{-1}$ bar$^{-1}$). Although having a kinetic diameter similar to isopropanol, THF exhibited high permeance (51.5 L m$^{-2}$ h$^{-1}$ bar$^{-1}$), because of its low viscosity. Importantly, this membrane also shows hexane and toluene permeances of 65.3 and 35.5 L m$^{-2}$ h$^{-1}$ bar$^{-1}$, respectively, which is much higher than the conventional polyamide membranes. For most of the solvents, we noticed that the product of permeance and viscosity is almost constant. Therefore, the solvent transport in MPCM membranes should

predominantly follow the pore-flow model, as a result of the permanently interconnected nano-isoporosity presence[34,35]. But the solvent size alone plays an important role in the permeation process. Water, with the smallest molecular size (0.38 nm), had higher permeance than toluene, though it has a higher viscosity.

**Chiral separation**. An important advantage of the membrane is that the homochirality of the trianglamine structure favors chiral selectivity. As shown in Supplementary Figs. 28 and 29, several amino acids have been tested for chiral separation. Their sizes are all below the membrane cut-off and a high rejection was not expected, but rather a distinction between levorotary and dextrorotary molecules. Separation experiments were conducted on both scenarios: feed solutions containing a single enantiomer with either L- or D-amino acid and racemic solutions with equal concentrations of both L- and D- enantiomers of amino acids: Valine, Leucine, Phenylalanine, and Tryptophan. Compared to the levorotary analogs, the membranes clearly rejected more the dextrorotary molecules, both in experiments with separated enantiomers and with racemic mixtures. The only exception was in the racemic of Valine, for which practically no separation was observed. In the case of single chiral enantiomer separation, the rejection of D-Leucine is 32%, while that of L-Leucine is only 3%. (Supplementary Fig. 28). For racemic separation, the rejection of D-Leucine is 31% compared to 17% rejection for L-Leucine (Supplementary Fig. 29). For Tryptophan, which is the tested molecule with size closest to the membrane cut-off, the results were even more encouraging. The membrane rejected around 24% of D-Tryptophan and near only 1% of L-Tryptophan, when testing a racemic mixture. These results indicate the perspective application of the membrane for superior chiral separation. The chiral selectivity of this membrane is assumed to be derived from the asymmetric environment of trianglamine for guest molecules through interactions such as hydrogen bonding, π–π stacking, and CH–π interactions[36].

**Molecular modeling of MPCM**. To better understand and correlate the performance of the MPCM with the microstructure and free voids for permeation, a molecular simulation was performed. A realistic structural model was generated using the Polymatic program and the corresponding properties were analyzed. The polymeric film was considered amorphous. Details of the simulations are given in the Supplementary Information and the results are shown in Fig. 5 and Supplementary Fig. 30. The model representation for MPCM in Fig. 5a shows the structure of the cross-linked trianglamine synthesized in this work as a distinct rigid network with a density of 1.163 g cm⁻³. By inserting theoretical probes of 1 Å radius, the highlighted blue areas in Fig. 5a resulted, indicating a large fraction of interconnected free volume in the MPCM, accessible to the probes. Figure 5b represents the distribution of voids with different sizes (scale from 1.4 to 3.2 Å), each color corresponding to the largest probe radius that could be inserted. A simple plot of the voids size distribution can be easily derived from the simulation results, revealing that the most frequently present voids have an average size of 3.8 Å (Fig. 5c). The fraction of voids of different sizes that are interconnected in relation to those isolated is represented in Fig. 5d–f. For this simulation, probes with different radii were inserted: 0.85, 1.2, and 1.55 Å, respectively. When the smallest probe with a radius of 0.85 Å is inserted, the voids are highly interconnected (green) and the nanofilm has high porosity, indicating that small molecules would easily permeate through the membrane selective layer. As the probe radius increases, the fraction of interconnected voids decreases. When a theoretical probe radius of 1.55 Å is used,

disconnected voids (red) fully dominate. This means that although a significant fraction of voids radius 1.55 Å or larger might be present in the selective layer, they would not contribute to the free diffusion of molecules of this size. We interpret the simulation as being exclusively related to the diffusivity of the cross-linked trianglamine layer and how it contributes to enhancing the transport of small molecules over larger ones, promoting a large size selectivity. The simulation does not consider thermodynamic interactions between the permeant molecules and the membrane, which could potentially contribute to the solubility and overall permeance or could even partially swell the membrane, altering the absolute sizes of cavities and pores. Nevertheless, the simulation reveals the high free volume and porosity of the MPCM, proving the significance of the macrocycle building blocks in creating permanently interconnected nano-porosity within the membranes for the diffusivity of permeants.

The modeling of the membrane prepared from TPC and trianglamine fragments was also conducted to verify the importance of the macrocycle architecture. The arrangement of the chemical backbone in Supplementary Fig. 30a is similar to that in Fig. 5a, but the fraction of void accommodating probes of 1 Å is clearly larger. Supplementary Fig. 30b indicated a pore size distribution similar to the MPCM membrane. What is most interesting is the difference in the interconnectivity of pores, comparing the two kinds of membranes. In opposite to what is observed for trianglamine membranes, the interconnectivity does not change much with the probe size in Supplementary Fig. 30d–f.

The MPCM membrane is highly selective, with the interconnectivity of pores promoting only the transport of molecules smaller than a specific size. The membrane prepared from trianglamine fragments is purely formed by physical entanglement, not cross-linked and the interconnectivity of pores accessible for permeation is less size-dependent. This might be a reason for the poorer selectivity of the fragment-based membrane compared to the trianglamine membrane. It rejects only 20% of congo red, while the trianglamine membrane is able to reject more than 90% of even smaller molecules.

## Discussion

A strategy for the fabrication of hyper-cross-linked thin-film composite membranes featuring an ultrathin film with interconnected microporosity was demonstrated. The unique structures were achieved by the interfacial polymerization of the macrocycle trianglamine. Due to the high molecular weight and ionized structure of trianglamine, a much slower diffusion took place during the interfacial polymerization, as well as a much lower solubility in the organic solution is observed, compared to the commonly used monomers such as m-phenylenediamine (MPD). Consequently, ultrathin selective layers (<10 nm) are formed with a high content of embedded trianglamine (62 wt%). Due to its intrinsic microporous structure, trianglamines can provide permanent channels for fast solvent permeance. Furthermore, because each trianglamine has multiple reacting sites for cross-linking, a hyper-cross-linked MPCM can be obtained, promoting high stability in harsh organic solvent environments, which is ideal for OSN applications. The MPCMs are characterized by exceptional permeance for both polar and apolar solvents that outperformed most of the commercial state-of-the-art OSN membranes in the market over a wide range of solvent polarity. Significantly, the high density of trianglamine unities in the membrane and the unique electrostatic property of the cavity, excellent shape, and charge selectivity are observed. The method is easily scalable for direct industrial production using current membrane production technologies. Furthermore, as the library of macrocycle molecules is vast and continuously increasing, the

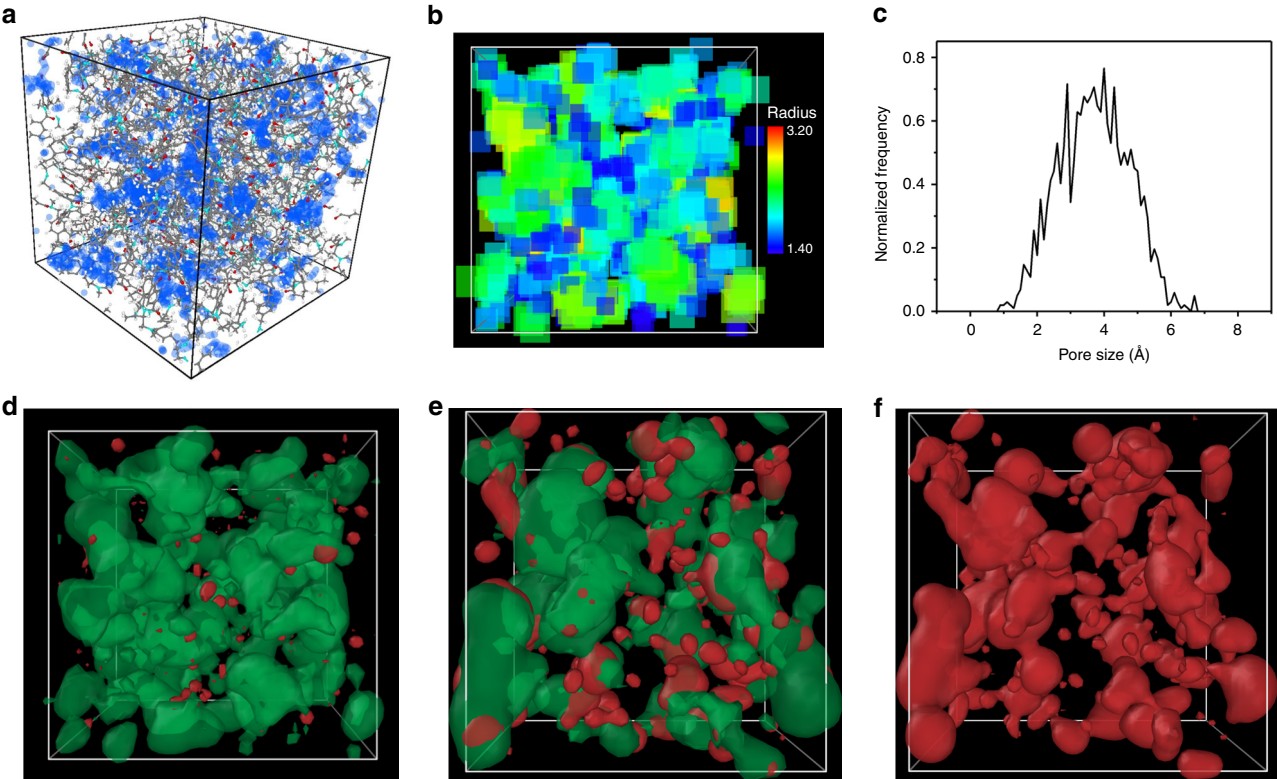

**Fig. 5 Molecular modeling of MPCM. a** Three-dimensional view of an amorphous cell of MPCM with a dimension of 30.39 Å × 30.39 Å × 30.39 Å, and the accessible surface at a probe radius of 1 Å marked blue. **b** Voids distribution with size distinguished by color. **c** Simulated pore size distributions of the constructed MPCM. Interconnected (green) and isolated (red) voids space considering probes of **d** 0.85 Å, **e** 1.2 Å, and **f** 1.55 Å radius, respectively.

tunability of related membranes is expected to further increase. This work will inspire further development of macrocycle-based polymeric membranes with ultrathin and microporous structure by rational design of the molecular building blocks for high performance and energy-efficient separations.

## Methods

**Materials**. Methanol, ethanol, acetone, hexane, and tetrahydrofuran (THF) were purchased from VWR Chemicals. Toluene and isopropanol (IPA) were procured from Sigma-Aldrich. Terephthaloyl chloride (TPC) was procured from TCI chemistry. All chemicals were used as received without further purification. The porous AAO support (Anodisc tm25, pore size 0.02 μm) was obtained from GE Healthcare Life Sciences. The porous PAN membrane was obtained from GMT GmbH. Deionized (DI) water (>18 MΩ cm) used in all experiments was filtered through a Millipore Milli-Q water purification system.

**Preparation of the trianglamine aqueous phase solution**. The synthesis of trianglamine was carried out as previously reported[31]. To prepare the aqueous phase solution for the interfacial polymerization, trianglamine was first added to water (10 ml) and sonicated to form turbid dispersion, followed by HCl (0.1 M) adding dropwise until the pH = 6.6 and the solution became almost transparent. After filtration with a syringe filter (0.22 μm of pore size), the clear solution was stored in a refrigerator for use and characterizations.

**Membrane preparation**. Hyper-cross-linked thin-film composite membranes were fabricated on the PAN ultrafiltration membrane (GMT GmbH, Rheinfelden, Germany) as supports via interfacial polymerization. First, the supports with the proper area were fixed with PTFE frames. 12 ml of aqueous solution (2% or 1% w/w) was added and the supports were impregnated for 10 min. The excess solution was removed and the supports surface was wiped with a rubber roller. Then the saturated supports were immersed in the TPC solution in hexane with 0.1% w/v concentration (unless otherwise stated) for a certain time, which resulted in the formation of an MPCM nanofilm on the top of the PAN supports. Finally, the resulting membrane's surface was rinsed with 20 ml of hexane to remove unreacted TPC. All membrane preparation experiments were carried out at room temperature and relative humidity of 60%.

Freestanding MPCM nanofilms were formed at the free interface between aqueous and organic solution using the same parameters as that of thin-film composite

membranes. With a certain period of time, the generated nanofilms can be deposited on silica wafers or alumina supports following the process depicted in Supplementary Fig. 8.

**Spectroscopic and thermal characterization**. Attenuated Total Reflection-Fourier transform infrared (ATR-FTIR) spectra were obtained on a Nicolet iS10 spectrometer with 16 scans and a resolution of 4 cm$^{-1}$. To get a sufficient amount of material for characterization, the membrane was continuously isolated from the free interface between the aqueous and organic solutions by using a silica wafer as a collector. Then the collected membrane material was washed with water and ethanol for one hour, respectively, and dried at 40 °C under vacuum. X-ray photoelectron spectroscopy (XPS) was performed on an Axis-Ultra DLD spectrometer using Al Kα radiation ($h\nu = 1486.6$ eV) under the base pressure of $3 \times 10^{-9}$ mbar. The binding energy data were calibrated in terms of the C 1s signal of aromatic carbon 284.5 eV was used as a reference. Calculations are detailed in the Supplementary Information.

Thermogravimetric analysis (TGA) was performed on the TA-Q500 with a temperature ramp of 5 °C min$^{-1}$ and a 20 ml min$^{-1}$ nitrogen flow rate.

**Morphological characterization**. Scanning electron microscopy (SEM) images were taken from a Zeiss Merlin field-emission scanning electron microscope at 3 kV and 100 pA with a working distance of 3 mm. High-resolution mode and Inlense detector were adopted. The samples were fixed on the specimen holders and then sputtered with 3 nm iridium in a Quorum Q150T to ensure conductivity. For cross-sectional SEM imaging, the membrane sample was cryogenically fractured in liquid nitrogen. Atomic force microscopy (AFM) images were obtained on a Dimension ICON scanning probe microscope under tapping mode. Surface morphology images and height profiles were obtained from silica wafer supported film samples at room temperature using FESPA etched silicon probes (spring constant = 2.8 N m$^{-1}$) with a scan rate of 1 Hz. Transmission Electron Microscope (TEM) images were collected on FEI Titan CT microscope operating at 300 kV. The samples were prepared as follows: the membranes were first immersed in an epoxy resin and cured at 65 °C, then thin sections having approximately 100 nm thickness were cut with Ultramicrotome Leica UC7 and collected on 300 mesh copper grids for TEM imaging. Wide-angle X-ray diffraction (XRD) patterns were collected on a Bruker D8 Advance with Cu KR radiation ($\lambda = 1.5406$ Å). The scan range is from 5 to 90° with a scan speed of 5° min$^{-1}$.

**OSN performance test**. A dead-end apparatus (Sterlitech stainless steel cells, HP4750) was used to carry out the nanofiltration experiments with the feed volume of 200 ml. The effective separation area was 13.8 cm². Prior to the test, the membrane was placed in the cell and compacted with both pure water and methanol for 1 h, respectively, to wash away any possible unreacted monomers and make the membrane stabilized. For pure solvents filtration, the operating transmembrane pressure was applied at 0.5–4 bar, based on the permeance of each solvent. Pure organic solvent permeance was measured by weighing the permeated solvent every 10 min under a steady state. The membrane selectivity was tested in solute-separation experiments using a series of dye methanol solutions with a concentration of 10–20 ppm at a transmembrane pressure of 1 bar unless otherwise stated. In order to exclude the effect of solute adsorption, both permeance and rejection were collected when a steady permeate was achieved. To investigate the influence of the transmembrane pressure on the permeate flux, experiments were performed under different pressures varying from 1 to 10 bar. The flux was measured by weighing the permeate every 10 min under a steady state. All nanofiltration experiments were performed at room temperature and repeated three times in parallel.

The solvent flux ($J$) is determined by Eq. 1, where $\Delta w$ is the weight of permeate collected during filtration time $\Delta t$, A is the effective membrane area of the cell, $\rho$ is the density of permeate, respectively.

$$J = \Delta w / \rho A \Delta t. \tag{1}$$

Permeance is determined by Eq. 2, where $\Delta P$ is the applied transmembrane pressure for the filtration experiment.

$$P = J / \Delta P. \tag{2}$$

Rejection was calculated from the concentration of feed and permeate solution using Eq. 3, where $C_f$ and $C_p$ represent the concentration of feed solution and permeate, respectively. NanoDrop UV–vis spectrophotometer was used to determine the concentration of feed and filtrate dye solution, using quartz cuvettes for sampling.

$$R = \left(1 - \frac{C_p}{C_f}\right) \times 100\%. \tag{3}$$

**Chiral separation**. The chiral selectivity was tested for different kinds of amino acids. The filtration of optically pure amino acid enantiomers solution (L-Valine (50 mg ml⁻¹), D-Valine (50 mg ml⁻¹), D-Leucine (10 mg ml⁻¹), L-Leucine (10 mg ml⁻¹), D-Phenylalanine (20 mg ml⁻¹), L-Phenylalanine (20 mg ml⁻¹), D-Tryptophan (10 mg ml⁻¹), and L-Tryptophan (10 mg ml⁻¹)) was conducted using a hand extruder Genizer (effective membrane area of 1.13 cm²). The concentration of permeate was analyzed on Rudolph polarimeter Autopol V for 2 s and repeated five times, with a laser wavelength of 365 nm. For the racemic separation experiment, the feed solution contained an equal concentration of L- and D-amino acid (50% L-amino acid: 50% D-amino acid): DL-Valine (25 mg ml⁻¹), DL-Leucine (5 mg ml⁻¹), DL-Phenylalanine (10 mg ml⁻¹), and DL-Tryptophan (2 mg ml⁻¹). The concentration of the permeate was analyzed using a Rudolph polarimeter Autopol V and UV–Vis absorbance equipment. All experiments were performed at room temperature and repeated three times in parallel.

## Data availability

The authors declare that the data supporting the findings of this study are available within the paper [and its Supplementary Information files]. Any additional detail can be requested from the corresponding authors.

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

## Acknowledgements

This research was supported by the King Abdullah University of Science and Technology (KAUST) base lines and CCF grant of the Advanced Membrane and Porous Materials Center. The TOC entry graph was created by Ivan Gromicho, Scientific Illustrator at KAUST. We thank Valentina-Elena Musteata, KAUST for the TEM characterization, and Kecheng Xie, China University of Mining Technology, for the molecular modeling.

## Author contributions

S.C., M.B., N.M.K., and S.N. conceived the project. T.H. designed the experiments. T.H. and M.B. carried out the materials synthesis. T.H. and M.B. performed materials characterization. T.H. and J.L carried out the nanofiltration measurements. P.H. and M.Y. carried out chiral separation experiments. T.H., S.C., and S.N. analyzed the data. G.Z. conducted molecular modeling. T.H., S.N., and N.M.K. wrote the manuscript. All authors discussed the results and commented on the manuscript.

## Competing interests

The authors declare no competing interests.
