## [Peer Review File · Nature Communications]

REVIEWER COMMENTS

Reviewer #1 (Remarks to the Author):

General comments

In this study, a molecularly porous cross-linked membrane (MPCM) was fabricated using triethylamines and terephthaloyl chloride (TPC) via interfacial polymerization for OSN. Several characterizations as well as molecular simulation were conducted. However, some claims are not well-supported. The following questions need to be satisfactorily addressed before publication in the journal.

Specific comments

- Pg 8, line 172 or Fig. 3a, the finger print region of triethylamine and the network actually differs quite a lot. What is reason for that?
- Please add error bars in the figures (i.e., Fig.2f; Fig.4a,4b,4e,4f; Fig.S21 etc.).
- In supplementary Figure 4, apparently, some pores are observed on the surface of MPCM/PAN thin-film composite membranes with 2% triethylamine in the aqueous phase. Are those defects? What is the repeatability of obtaining defect free membranes from the IP process?
- It was claimed that “each triethylamine is covalently linked with other 4 triethylamines via TPC bridging”. Is there any possibility for two acyl chloride groups of one TPC react with the same triethylamine?
- Is there any evidence to indicate that the rejection of MPCM/PAN thin-film composite membranes is contributed by molecular sieve rather than adsorption? Please increase the dye concentration (10-20 ppm is quite low) and feed volume for more reliable filtration test results.
- In general, 24-hour for OSN filtration test seems cannot define as a long-term, to prove the long-term stability of formed membranes please extend the testing duration. Please also provide long term separation performance.
- The authors claimed that the membrane is charge-selective for molecules with similar size from the separation results of MO and NR. A positive dye with similar molecular weight, methylene blue (320 g mol⁻¹), should also be tested to validate this claim.
- Please explain why this membrane favors chiral selectivity, especially all L-enantiomer shows higher rejection than R-enantiomer?
- Pg 13, Is there any validation of the pore size distribution from Molecular Simulation? Since there are intrinsic pores, shouldn't there be a bimodal distributions of the pores?
- Pg 14, line 343, what does “as for polymeric gas separation membranes” mean? The permeability of gas separation membranes is usually determined by solubility and diffusivity.
- Pg 15, please state clearly the nanofiltration experimental conditions such as the experimental duration, feed volume, filtered volume and the exact pressure rather than a range of as 0.5-10 bar for the various experiments.

Reviewer #2 (Remarks to the Author):

The authors report on an original and relevant extension of the state-of-the-art in the fabrication of high performance separation membrane for challenging separations. The key innovation is the successful utilization of a building block for intrinsically microporous polyamides during the well established and scalable interfacial polymerization to yield thin-film composite membranes. The so called „trianglamine“ (before by others used to obtain porous organic frameworks) was used as special monomer, leading to highly crosslinked and ultrathin polyamide films, apparently without major defects even at very low thickness. All results indicate that the barrier properties of the resulting membranes are largely different from those with conventional cross-linked semiaromatic polyamides, high fluxes for different organic solvents and water are obtained at a cut-off below 500 g/mol.

Based on the comprehensive chemical characterization, it is highly likely that this can be attributed to an ultramicroporosity created by a hyper-crosslinked chemical network with the „void“-creating trianglamine. A more direct characterization of the pore structure of the materials, using adsorption isotherms or positron annihilation life time spectroscopy, is admittedly challenging; that this has not been done is a weakness of the work, but the reviewer can accept that such experiments had not been attempted here. One control experiment with a „linear fragment“, leading to „practically the same chemistry“ has been performed, leading to different reactivity and membranes with conventional properties and separation performance. This alone, however, provides rather little mechanistic insights.

The authors put more emphasis on another „proof“ for the special microporous structure, obtained by „molecular modelling“ (p. 13-14). However, the depth of the description, provided only in SI, is insufficient to judge about the validity of the results! Furthermore, they should also include modelling using the same methodology for the polymer obtained with the „linear fragment“ (cf. above), and they should then provide a quantitative comparison of the main „output“ (data in Fig. 5), obtained with „an inserted probe“ (they completely forgot to write how that analysis had been performed!) between the conventional („control“) and the novel „microporous“ polymer. This would strengthen the scientific impact.

In addition, there are indications that the particular chemical structure of the building block has also influence on the barrier properties.

On the one hand, the authors suggest that the „electron-rich“ interior of the building block „could prevent netatively charged molecules from going through it“ (p. 11). Again, the details of the „simulation“ are missing; only a result is shown in SI. However, I would expect more concrete empirical (experiments with varied solutes in different solvents) and fundamental considerations to support this „claim“ (that is also made in the Abstract).

On the other hand, the additional „important advantage“ of the building block, its chirality with possible influence on separation selectivity, is also probed in filtration experiments. Unfortunately, the information provided is not sufficient to judge about the evoked enantio-selectivity. The text in the Experimental („D/L-valine“ etc.; p. 16) may suggest that racemates of amino acids had been used during filtration (I would very strongly encourage that, but am not sure because in main text they report rejections for the individual isomers and also they only talk about „concentration“; see next!).

The further analysis is unclear (cf. p. 16-7): „concentration of the permeate was determined on ... polarimeter for two seconds and repeated five times.“ Does this mean that enantiomeric excess of permeate was analyzed? Or did they use polarimetry for measuring just the concentration of substances that could be quantified also easier with other methods? And what is the meaning of „two seconds“ and „repeated five times“?

Those are not just formal questions, but topics important for a discussion about the reason of the evoked chiral selectivity (that discussion is missing in the paper). Is the preferential binding in the barrier layer responsible for the different „rejection“ values? If yes, „rejection“ will change after all sites had been saturated, i.e. steady state-data must be acquired and reported. If no difference between D- and L-aminoacids is observed in single solute experiments, there is still a chance that this may be different for mixtures. If such steady-state single solute „rejection“ values would be different for D- and L-aminoacids, the influence of solute concentration and flux on rejection must be analyzed as well, and based on all results the mechanism of transport selectivity can be well discussed based on established models (e.g. „fixed carrier“).

Finally, the language and (thus) formal correctness of the paper must be carefully revised; the text is full with small errors (mainly grammar; e.g. inconsistency between singular and plural within a sentence or mixing up the concepts „adjective“ vs. „adverb“; also occasional wrong words), but also use of „lab slang“ (e.g.: p. 5: „synthesis feasibility (> 90%)“ /what is meant here?;/ p. 8: „steric effect of the reaction“). In particular, the SI file is still more like a draft. In „Supplementary Table 3“, the unit for permeance is missing; and in the names for the samples from the own work, there is an error (same acronym /“MPCN/PAN(2%,10s)“/ used twice; I assume the second should be „1%“). Such weakness should be avoided when submitting to a high-rank Journal.

Overall, publication in the Journal may be possible after major revisions addressing above points.

Reviewer #3 (Remarks to the Author):

The manuscript describes the fabrication and extensive characterization of molecularly porous cross-linked membranes synthesized by interfacial polymerization using a novel triethylamine monomer. The performance of the membranes are shown to be in the nanofiltration range, with extremely small thickness down to 3.5 nm, very high flux and even the capacity of distinguishing chiral isomers. This is an excellent work, with the potential of opening new avenues for tunable thin film composite nanofiltration membranes that can be used in a wide range of media, including harsh organic solvents. The work is thoroughly conducted and carefully and clearly discussed in the manuscript. I recommend its publication. The following are a few points which may help further clarify the text.

Page 5, line 6: What is meant by 90% synthesis feasibility? Is it the yield? This should be more clearly expressed.

Page 8, line 3: “The hydrophilicity of MPCM was confirmed by the water contact angles (Supplementary Fig. 17).” It would be better to state this as the hydrophobicity of MPCM as the contact angles appear to be close to 90 degrees.

Page 12, paragraph starting on line 6: It is stated that the solvent permeance through the membranes scales with viscosity and not solvent size. This implies a convective flow mechanism through the triethylamine openings rather than solution-diffusion through the whole film, which is expected. Can the authors elaborate a bit more on this transport mechanism perhaps theoretically relating the permeance to the pore size and membrane thickness?

Answer point by point

REVIEWER COMMENTS

Reviewer #1 (Remarks to the Author):

General comments

In this study, a molecularly porous cross-linked membrane (MPCM) was fabricated using triethylamines and terephthaloyl chloride (TPC) via interfacial polymerization for OSN. Several characterizations as well as molecular simulation were conducted. However, some claims are not well-supported. The following questions need to be satisfactorily addressed before publication in the journal.

Specific comments

- Pg 8, line 172 or Fig. 3a, the finger print region of triethylamine and the network actually differs quite a lot. What is reason for that?

Response: We expanded the spectra to clarify and support that most peaks in the fingerprint region can be seen in the spectra of triethylamine and the network. Indeed, some of the relative intensities change, mainly increasing at around 750, 1260, and 1400 cm^{-1} for the network. The first is relative to the O=C-N bending in-plane in amides, while in amines N-H asymmetric bend is strong at lower wavenumber, also with C-N-C at around 427 cm^{-1} . 1260 is probably relative to C-N stretching. 1400 cm^{-1} is known to be present in amides, but the absorption is less specific. We revised our explanation in the revised text (highlighted page 8).

- Please add error bars in the figures (i.e., Fig.2f; Fig.4a,4b,4e,4f; Fig.S21 etc.).

Response: Error bars have been added where needed as requested by the reviewer.

- In supplementary Figure 4, apparently, some pores are observed on the surface of MPCM/PAN thin-film composite membranes with 2% triethylamine in the aqueous phase. Are those defects? What is the repeatability of obtaining defect free membranes from the IP process?

Response: We thank the reviewer for this important comment. It gives us the chance to clarify even more the information. The membranes are dense without defects. A clear indication is the high rejection obtained for small molecules in organic solvent in Figure 4b, which is relative to the 2% triethylamine membranes. We should have better specify in the caption. All error bars in Fig. 4 represent standard deviation for at least three independent measurements. All membranes prepared with 2% triethylamine had good separation performance tested with dye molecules. For example, for the membrane of 2% triethylamine and reaction time 10min, the rejection of acid fuchsin recorded for different membranes was 98.3%, 100%, 100%; the rejection of methyl orange was 76.9%, 79.9%, 75.3%; the rejection of orange G was 98.5%, 97.6%, 95.1%; for reaction time of 5min, the rejection of acid fuchsin was 98.7%, 98.3%, 97.9%. This high repeatability confirms that the membranes are defect free. When the IP layer is very thin, it tends to cover the porous sublayer following the morphology of the support. What we see are not pores, but the very thin, relatively flexible and dense film deposited on the porous sub-surface with its characteristic relief.

- It was claimed that “each triethylamine is covalently linked with other 4 triethylamines via TPC bridging”. Is there any possibility for two acyl chloride groups of one TPC react with the same triethylamine?

Response: We believe that the possibility that two acyl chloride groups of one TPC reacting with the same triethylamine would be low due to steric hindrance. Kuhnert et al. (*Tetrahedron Lett.* **2006**, 47, 6915) reports a reaction between triethylamine and simple alkyl dihalides. But the aromatic TPC is much bulkier and less flexible.

See below how it would in principle occur:

It would be very difficult due to the rigidity and planarity of the amide bond and the TPC (especially tertiary amide) in addition to the chirality between the two neighboring nitrogen atoms attached to the cyclohexane (*Chem. Eur. J.* **2006**, *12*, 1807 – 1817, *Tetrahedron* **2003**, *59*, 9323–9331). Moreover, IP was done at room temperature and changing the conformation is not favored enough to change the orientation of the cavity of the macrocycle. Consequently, the environment of the macrocycle will be crowded after the first covalent reaction. This is what we meant by “steric”. A second attack to form a covalent bond will be sterically hindered by the crowded environment and therefore it is kinetically more favorable to attack an “open space” on a new ring.

- Is there any evidence to indicate that the rejection of MPCM/PAN thin-film composite membranes is contributed by molecular sieve rather than adsorption? Please increase the dye concentration (10-20 ppm is quite low) and feed volume for more reliable filtration test results.

Response: This is certainly an important point. First, we tested the dye rejection of the plain PAN support, and this was extremely low. Therefore, the adsorption on PAN would not be significant. Now for the composite membrane, in order to exclude the effect of solute adsorption on the formed network, both permeance and rejection were collected when a steady permeate was achieved. To further confirm it, long term (> 48h) separation of dye solution (congo red) with high concentration (100 ppm) has also been conducted as suggested. High rejection (99%) was obtained as well. The retentate is highly concentrated compared to the feed solution, closing the mass balance, without indication of adsorption on the thin-film surface. The result and discussion have been added in revised manuscript as well as supplementary information.

- In general, 24-hour for OSN filtration test seems cannot define as a long-term, to prove the long-term stability of formed membranes please extend the testing duration. Please also provide long term separation performance.

Response: Per the reviewer’s suggestion, long term pure solvent filtration of more than 48 h was conducted and the data has been added to the revised supplementary information. Long term separation of dye solution (congo red) with high concentration (100 ppm) has been conducted as well. High rejection (99%) was observed. The result has been added to the manuscript and supplementary information, as suggested.

- The authors claimed that the membrane is charge-selective for molecules with similar size from the separation results of MO and NR. A positive dye with similar molecular weight, methylene blue (320 g mol⁻¹), should also be tested to validate this claim.

Response: This is a good suggestion. The separation performance for positive dyes (methylene blue) has been tested and 36% of rejection was obtained, which confirmed that the membrane is charge-selective. We have added the result to the revised manuscript and supplementary information.

- Please explain why this membrane favors chiral selectivity, especially all L-enantiomer shows higher rejection than R-enantiomer?

Response: The results of the chiral separation experiments in Supplementary Figure 28 indicated higher retention of dextrorotatory molecules compared to levorotatory. Based on the literature (Tanaka, K.; Fukuda, N.; Fujiwara, T. Trianglamine as a new chiral shift reagent for secondary alcohols. *Tetrahedron: Asymmetry* 2007, 18, 2657) for single trianglamine molecules, the chiral selectivity of this membrane is probably derived from the asymmetric environment of the trianglamine for various guest molecules through interactions such as hydrogen bonding, π - π stacking, and CH- π interactions. Moreover, the stereospecific behavior could be explained by symmetry constraints due to spin polarization (Kumar, A. et al. Chirality-induced spin polarization places symmetry constraints on biomolecular interactions. *PNAS* 114, 2474-2478, (2017) and favorable interactions between molecules with similar spatial configuration. Discussion on chiral separation has been added in the revised manuscript.

- Pg 13, Is there any validation of the pore size distribution from Molecular Simulation? Since there are intrinsic pores, shouldn't there be a bimodal distributions of the pores?

Response: The direct measurement of the pore size distribution at this low range is challenging. This is far below the range of the porometer we have in our institution. A method like PALS could give an indirect information, but PALS for thin layers is available in just a few labs in the world and gives an indirect information. The modeling is a tool to better understand our results. What we could see from the modeling is that we do not have a bimodal distribution in the classical sense, but we have interconnected and non-connected pores. The relation between them changes with the monomer we use, if trianglamine or a fragment of trianglamine. We extended the modeling to better explain our results in the revised manuscript.

- Pg 14, line 343, what does "as for polymeric gas separation membranes" mean? The permeability of gas separation membranes is usually determined by solubility and diffusivity.

Response: This is exactly what we mean. Not only for gas, but also in the case of dense selective membranes, as in the case reported here, the separation could be a combination of solubility and diffusivity. More free volume contributes to the diffusivity. Interaction with different chemical groups in

the layer contribute to the solubility. Both contributions will lead to the final value of the measured membrane permeance, as well as the layer thickness. This is now clarified in the revised text.

- Pg 15, please state clearly the nanofiltration experimental conditions such as the experimental duration, feed volume, filtered volume and the exact pressure rather than a range of as 0.5-10 bar for the various experiments.

Response: More details about the nanofiltration experimental conditions, including experimental duration, feed volume, exact pressure, have been added in the revised manuscript as suggested.

Reviewer #2 (Remarks to the Author):

The authors report on an original and relevant extension of the state-of-the-art in the fabrication of high performance separation membrane for challenging separations. The key innovation is the successful utilization of a building block for intrinsically microporous polyamides during the well established and scalable interfacial polymerization to yield thin-film composite membranes. The so called „trianglamine“ (before by others used to obtain porous organic frameworks) was used as special monomer, leading to highly crosslinked and ultrathin polyamide films, apparently without major defects even at very low thickness. All results indicate that the barrier properties of the resulting membranes are largely different from those with conventional cross-linked semiaromatic polyamides, high fluxes for different organic solvents and water are obtained at a cut-off below 500 g/mol.

Based on the comprehensive chemical characterization, it is highly likely that this can be attributed to an ultramicroporosity created by a hyper-crosslinked chemical network with the „void“-creating trianglamine. A more direct characterization of the pore structure of the materials, using adsorption isotherms or positron annihilation life time spectroscopy, is admittedly challenging; that this has not been done is a weakness of the work, but the reviewer can accept that such experiments had not been attempted here. One control experiment with a „linear fragment“, leading to „practically the same chemistry“ has been performed, leading to different reactivity and membranes with conventional properties and separation performance. This alone, however, provides rather little mechanistic insights.

The authors put more emphasis on another „proof“ for the special microporous structure, obtained by „molecular modelling“ (p. 13-14). However, the depth of the description, provided only in SI, is insufficient to judge about the validity of the results! Furthermore, they should also include modelling using the same methodology for the polymer obtained with the „linear fragment“ (cf. above), and they should then provide a quantitative comparison of the main „output“ (data in Fig. 5), obtained with „an inserted probe“ (they completely forgot to write how that analysis had been performed!) between the conventional („control“) and the novel „microporous“ polymer. This would strengthen the scientific impact.

Response: We thank the reviewer for this comment and understand that the description of the modelling part was not detailed enough. We have now significantly extended our explanation in the revised manuscript. It is an excellent suggestion to add modelling under comparable conditions for the membrane prepared with fragments of trianglamine. This has been conducted and added to the new version.

Moreover, the direct experimental validation of the pore size distribution in the range we have in this membrane is of course also an excellent suggestion but highly challenging. The pore size is below the range detectable by the porometer we have. PALS could give an indirect estimation, but PALS for thin layers is available in selected labs in the world. Using adsorption isotherms, like BET, would be a good idea, in principle. But it would make sense only to measure it for the thin layer alone, detached from the porous support, otherwise the influence of the much more porous support would be too big. We believe that the recently added modeling experiments, per the reviewers suggestion, clearly supports our finding and conclusions at this stage (page 15).

In addition, there are indications that the particular chemical structure of the building block has also influence on the barrier properties.

On the one hand, the authors suggest that the „electron-rich“ interior of the building block „could prevent negatively charged molecules from going through it“ (p. 11). Again, the details of the „simulation“ are missing; only a result is shown in SI. However, I would expect more concrete empirical (experiments with varied solutes in different solvents) and fundamental considerations to support this „claim“ (that is also made in the Abstract).

Response: Thank you very much for the comments. We are very sorry that we forgot to put the details of triethylamine molecular simulation into the manuscript. In the revised manuscript, details of the simulation have been added.

On the other hand, the additional „important advantage“ of the building block, its chirality with possible influence on separation selectivity, is also probed in filtration experiments. Unfortunately, the information provided is not sufficient to judge about the evoked enantio-selectivity. The text in the Experimental („D/L-valine“ etc.; p. 16) may suggest that racemates of amino acids had been used during filtration (I would very strongly encourage that, but am not sure because in main text they report rejections for the individual isomers and also they only talk about „concentration“; see next!).

Response: We agree we should have detailed better the description of the chiral separation experiments. The experiments reported in the submitted version were carried out with optically pure solutions and not on racemic ones: L-Valine (50 mg/ml), D-Valine (50 mg/ml), D-Leucine (10 mg/ml), L-Leucine (10 mg/ml), D-Phenylalanine (20mg/ml), L-Phenylalanine (20mg/ml), D-Tryptophan (10 m/ml), L-Tryptophan (10 m/ml). We have modified the description of the chiral separation experiment in the revised manuscript. Now we added also results conducted with the racemic solution in the revised version (page 13).

The further analysis is unclear (cf. p. 16-7): „concentration of the permeate was determined on ... polarimeter for two seconds and repeated five times.“ Does this mean that enantiomeric excess of permeate was analyzed? Or did they use polarimetry for measuring just the concentration of substances that could be quantified also easier with other methods? And what is the meaning of „two seconds“ and „repeated five times“?

Response: Since we only used optically pure amino acid solutions for testing in the previous version of the manuscript, the data reported was not the enantiomeric excess. Polarimetry was used to measure the

concentration of amino acid solution. Compared to other techniques suitable for amino acid concentration measurement, polarimetry is relatively easy to handle. Each chiral separation experiment was repeated three times on three separate membranes. During each separation, the concentration of the permeate was analyzed using Rudolph polarimeter Autopol V. The polarimeter measures the optical solution at 25°C using 365 nm laser for 2 seconds and repeat this process five times to measure the repeatability of the reading. In order to avoid unnecessary confusion, we delete “for two seconds and repeated five times” from the manuscript. We added results for the racemic too in the new version (page 13).

Those are not just formal questions, but topics important for a discussion about the reason of the evoked chiral selectivity (that discussion is missing in the paper). Is the preferential binding in the barrier layer responsible for the different „rejection“ values? If yes, „rejection“ will change after all sites had been saturated, i.e. steady state-data must be acquired and reported. If no difference between D- and L-aminoacids is observed in single solute experiments, there is still a chance that this may be different for mixtures. If such steady-state single solute „rejection“ values would be different for D- and L-aminoacids, the influence of solute concentration and flux on rejection must be analyzed as well, and based on all results the mechanism of transport selectivity can be well discussed based on established models (e.g. „fixed carrier“).

Response: We thank the reviewer for this important point. We extended the discussion in the revised version. Based on the literature, the chiral selectivity of this membrane is assumed to be derived from the asymmetric environment of triethylamine for various guest molecules through interactions such as hydrogen bonding, π - π stacking, and CH- π interactions. (Tanaka, K.; Fukuda, N.; Fujiwara, T. *Triethylamine as a new chiral shift reagent for secondary alcohols*. *Tetrahedron: Asymmetry* 2007, 18, 2657). It is encouraging to continue to investigate the chirality of this system in detail in the future.

Finally, the language and (thus) formal correctness of the paper must be carefully revised; the text is full with small errors (mainly grammar; e.g. inconsistency between singular and plural within a sentence or mixing up the concepts „adjective“ vs. „adverb“; also occasional wrong words), but also use of „lab slang“ (e.g.: p. 5: „synthesis feasibility (> 90%)“ /what is meant here?/; p. 8: „steric effect of the reaction“). In particular, the SI file is still more like a draft. In „Supplementary Table 3“, the unit for permeance is missing; and in the names for the samples from the own work, there is an error (same acronym /“MPCN/PAN(2%,10s)“/ used twice; I assume the second should be „1%“). Such weakness should be avoided when submitting to a high-rank Journal.

Response: Thank you very much for your comments. We are sorry about the grammar error and typos in our manuscript. We carefully went through our revised manuscript. The informal language use has also been modified. The ‘synthesis feasibility’ was substituted by “synthesis with yield higher than 90%” and ‘steric effect of the reaction’ have been also substituted by “steric hindrance”, which is commonly used in chemistry.

Overall, publication in the Journal may be possible after major revisions addressing above points.

Reviewer #3 (Remarks to the Author):

The manuscript describes the fabrication and extensive characterization of molecularly porous cross-linked membranes synthesized by interfacial polymerization using a novel triethylamine monomer. The performance of the membranes are shown to be in the nanofiltration range, with extremely small thickness down to 3.5 nm, very high flux and even the capacity of distinguishing chiral isomers. This is an excellent work, with the potential of opening new avenues for tunable thin film composite nanofiltration membranes that can be used in a wide range of media, including harsh organic solvents. The work is thoroughly conducted and carefully and clearly discussed in the manuscript. I recommend its publication. The following are a few points which may help further clarify the text.

Page 5, line 6: What is meant by 90% synthesis feasibility? Is it the yield? This should be more clearly expressed.

Response: We mean that the triethylamine is relatively easy to synthesize with a high yield ~ 90%. We substituted by "synthesis yield higher than 90%" in the revised manuscript.

Page 8, line 3: "The hydrophilicity of MPCM was confirmed by the water contact angles (Supplementary Fig. 17)." It would be better to state this as the hydrophobicity of MPCM as the contact angles appear to be close to 90 degrees.

Response: Thank you for the comment. The mentioned statement has been modified as suggested.

Page 12, paragraph starting on line 6: It is stated that the solvent permeance through the membranes scales with viscosity and not solvent size. This implies a convective flow mechanism through the triethylamine openings rather than solution-diffusion through the whole film, which is expected. Can the authors elaborate a bit more on this transport mechanism perhaps theoretically relating the permeance to the pore size and membrane thickness?

Response: For most of the solvents, we notice that the product of permeance and viscosity is almost a constant. Therefore, we think the solvent transport in MPCM membranes follows the pore-flow model (Yang, Q. et al. Ultrathin graphene-based membrane with precise molecular sieving and ultrafast solvent permeation. *Nat. Mater.* 16, 1198–1202 (2017); Mulder, M. *Basic Principles of Membrane Technology* (Kluwer Academic Publishers Group, Dordrecht, 1996).). The solvent size plays a more important role in the permeation process than solubility or interaction with chemical functionalities. For instance, water has the highest permeance, probably because of its smallest molecular size (0.38 nm), compared for instance with toluene. We extended and clarified the discussion in the revised manuscript.

REVIEWERS' COMMENTS

Reviewer #1 (Remarks to the Author):

The revised version is much better and can be acceptable.

Reviewer #2 (Remarks to the Author):

The authors have very well adressed all comments by the reviewers. In particular, the additional experimental data, the update of dicussion and the formal revisions are strenghening the paper. I recommend to accept it.

Reviewer #3 (Remarks to the Author):

I find the revised version of the manuscript suitable for publication.